# Impact of the COVID-19 Pandemic on Community Antibiotic Prescribing and Stewardship: A Qualitative Interview Study with General Practitioners in England

**DOI:** 10.3390/antibiotics10121531

**Published:** 2021-12-14

**Authors:** Aleksandra J. Borek, Katherine Maitland, Monsey McLeod, Anne Campbell, Benedict Hayhoe, Christopher C. Butler, Liz Morrell, Laurence S. J. Roope, Alison Holmes, Ann Sarah Walker, Sarah Tonkin-Crine

**Affiliations:** 1Nuffield Department of Primary Care Health Sciences, University of Oxford, Oxford OX2 6GG, UK; katherine.maitland@st-hughs.ox.ac.uk (K.M.); christopher.butler@phc.ox.ac.uk (C.C.B.); sarah.tonkin-crine@phc.ox.ac.uk (S.T.-C.); 2National Institute for Health Research (NIHR) Health Protection Research Unit in Healthcare Associated Infections and Antimicrobial Resistance, Imperial College London, London W12 0NN, UK; monsey.mcleod@nhs.net (M.M.); anne.campbell@imperial.ac.uk (A.C.); alison.holmes@imperial.ac.uk (A.H.); 3Centre for Medication Safety and Service Quality, Pharmacy Department, Imperial College Healthcare NHS Trust, London W6 8RF, UK; 4NIHR Imperial Patient Safety Translational Research Centre, Imperial College London, London W6 8RF, UK; 5Department of Primary Care and Public Health, School of Public Health, Imperial College London, London W6 8RP, UK; b.hayhoe@imperial.ac.uk; 6NIHR Applied Research Collaboration Northwest London, London W6 8RP, UK; 7Health Economics Research Centre, Nuffield Department of Population Health, University of Oxford, Oxford OX3 7LF, UK; liz.morrell@ndph.ox.ac.uk (L.M.); laurence.roope@dph.ox.ac.uk (L.S.J.R.); 8NIHR Oxford Biomedical Research Centre, Oxford OX3 9DU, UK; sarah.walker@ndm.ox.ac.uk; 9NIHR Health Protection Research Unit in Healthcare Associated Infections and Antimicrobial Resistance, University of Oxford, Oxford OX3 9DU, UK; 10Nuffield Department of Medicine, University of Oxford, Oxford OX3 9DU, UK

**Keywords:** antibiotics, antimicrobial stewardship, general practice, COVID-19, qualitative

## Abstract

The COVID-19 pandemic has had a profound impact on the delivery of primary care services. We aimed to identify general practitioners’ (GPs’) perceptions and experiences of how the COVID-19 pandemic influenced antibiotic prescribing and antimicrobial stewardship (AMS) in general practice in England. Twenty-four semi-structured interviews were conducted with 18 GPs at two time-points: autumn 2020 (14 interviews) and spring 2021 (10 interviews). Interviews were audio-recorded, transcribed and analysed thematically, taking a longitudinal approach. Participants reported a lower threshold for antibiotic prescribing (and fewer consultations) for respiratory infections and COVID-19 symptoms early in the pandemic, then returning to more usual (pre-pandemic) prescribing. They perceived the pandemic as having had less impact on antibiotic prescribing for urinary and skin infections. Participants perceived the changing ways of working and consulting (e.g., proportions of remote and in-person consultations) in addition to changing patient presentations and GP workloads as influencing the fluctuations in antibiotic prescribing. This was compounded by decreased engagement with, and priority of, AMS due to COVID-19-related urgent priorities. Re-engagement with AMS is needed, e.g., through reviving antibiotic prescribing feedback and targets/incentives. The pandemic disrupted, and required adaptations in, the usual ways of working and AMS. It is now important to identify opportunities, e.g., for re-organising ways of managing infections and AMS in the future.

## 1. Introduction

Antimicrobial stewardship (AMS), including optimising antibiotic prescribing and use, is a public health priority designed to mitigate the development and impact of antimicrobial resistance [1,2]. Although antibiotic prescribing in general practice and community settings has steadily decreased since 2014, antibiotic-resistant bloodstream infections have increased by 32% since 2015 [3]. Continued AMS efforts are needed as antimicrobial resistance is quickly becoming another health emergency.

The emergence of severe acute respiratory syndrome coronavirus 2 (SARS-CoV-2) and the resulting COVID-19 pandemic have posed new challenges to assessing and treating patients with respiratory tract infections (RTIs). International studies have shown that the majority of patients admitted to hospitals with COVID-19 were prescribed empirical antibiotics despite a low prevalence of bacterial co-infection and regardless of illness severity [4,5,6,7,8]. A UK-wide survey of hospital pharmacists (members of AMS teams or AMS leads) in June 2020 found that most respondents reported a reduction in AMS activity and a negative impact of the pandemic on AMS [9]. Consequently, concerns were raised about the impact of COVID-19 on antimicrobial prescribing, AMS and future consequences for antimicrobial resistance, health and the environment, with calls to keep AMS as a priority [10,11,12,13,14,15,16,17].

Studies from primary care in England have shown an overall decrease in antibiotic prescribing during 2020 compared to pre-pandemic [18,19,20,21,22,23], with similar trends identified in Scotland [24], Northern Ireland [25] and other countries [26,27,28,29]. However, one study on UK data noted an initial increase in antibiotic prescribing in March 2020 before prescribing levels dropped below those expected (with the lowest rate in May) [20]. Similarly, in Scotland the number of antibiotic prescriptions was 44% higher in the week of 22 March 2020 than in a corresponding week in 2019, followed by lower-than-expected prescribing during the national lockdown in April and May 2020 [24]. Moreover, one study showed that although antibiotic prescribing decreased between April and August 2020 by 15.5% compared to a similar period in 2019, the number of prescriptions was actually 6.7% higher than expected when taking into account the fewer appointments (−50% in-person and −20.8% all types of) [18]. This might suggest that suboptimal prescribing for patients unlikely to benefit from antibiotics persists and may have increased during the pandemic; alternatively, patients’ threshold for consulting might have increased and more may have required antibiotics. Different factors were proposed as potentially influencing changes in antibiotic prescribing during the pandemic, e.g., fewer face-to-face consultations, more remote consultations and lower infection transmission due to public health measures (e.g., social distancing) [13,14,18]. However, there is little empirical evidence regarding these factors and prescribers’ own experiences—a better understanding of antibiotic prescribing and AMS during the pandemic may help with AMS.

The aim of this qualitative interview study was to explore general practitioners’ (GPs) views on, and experiences of, the impact of the COVID-19 pandemic on antibiotic prescribing and AMS in English general practice.

## 2. Results

Twenty-four interviews were conducted with 18 GPs, lasting on average 36 (18–54) minutes. In autumn 2020 there were 14 interviews (eight GPs were from the on-going STEP-UP study and six were additionally recruited); in spring 2021 there were 10 interviews (including the same six additional GPs from autumn). After this we stopped recruitment as we considered the data sufficient to answer the research question and develop rich categories and themes supported by multiple data extracts from different participants (i.e., more interviews were considered unlikely to affect the main findings).

Below we present the findings, supported by empirical data (quotes). Additional quotes are in Appendix A. We first describe how the COVID-19 pandemic was perceived to impact antibiotic prescribing and then the perceived reasons for these impacts.

### 2.1. Fluctuations in Antibiotic Prescribing during COVID-19

Participants reported that their “threshold” for antibiotic prescribing for RTIs and COVID-19 symptoms was lower at the start of the COVID-19 pandemic compared to pre-pandemic, i.e., they used antibiotics more “*indiscriminately*” and their likelihood of prescribing empirical antibiotics increased. This was perceived to be influenced by the emergency situation and related factors (described in the following sections).
“*Right at the beginning, when COVID first struck, (…) we were treating a lot of chest infections virtually which sounds and goes against everything that we’ve ever been trained to do but we were told we absolutely mustn’t spread COVID, therefore we must keep our distance, that was when PPE was in very short supply. And talking to my colleagues about it, we certainly were prescribing more antibiotics.*”(GP02 October 2020)

Over time and as the situation changed, participants described a gradual return to a more “normal” (i.e., pre-pandemic) approach to antibiotic prescribing.
“*As the pandemic has moved on, we’ve gone a little bit more back to normal, I suppose, being more strict, in that we would expect patients to come in if there is a need for an antibiotic (…) it’s [antibiotic prescribing] probably gone back down to a more normal level.*”(GP14 May 2021)

However, as almost all participants had not seen the prescribing data in 2020 they were typically unsure as to whether their practices’ antibiotic prescribing rates had changed. Some guessed that the overall number of antibiotics prescribed might be higher due to the lower threshold for prescribing; others thought that it might be lower as a result of fewer consultations for RTIs.

Finally, most participants reported that antibiotic prescribing for urinary tract infections (UTIs) and skin infections was less affected by COVID-19 because they were seen as more easily managed remotely than RTIs. For UTIs, most participants described deciding about antibiotics based on reported symptoms (according to algorithms/guidelines) and/or urine samples dropped off at practices with “minimal contact”; some referred to being used to (or moving towards) prescribing for uncomplicated UTIs remotely already before the pandemic. For skin infections, they reported using photos or videos of the affected areas to help the assessment. Only a few participants thought that they were still more likely to prescribe antibiotics for these conditions in remote consultations than face-to-face.
“*If people meet the number of symptoms threshold that’s set in the [local] guidance, we would just prescribe for them. Most UTIs have generally been dealt with remotely anyway (…) that’s probably never really changed.*”(GP12 May 2021)
“*Skin infections, because the patient can now send photos of their skin lesion, COVID or no COVID that was happening anyway. Those are quite easy to diagnose remotely.*”(GP17 June 2021)

The fluctuations in antibiotic prescribing (mostly for RTIs) were influenced by the factors described below.

### 2.2. Changing Ways of Working and Consulting

Participants described the changes in their ways of working and consulting, triggered by the pandemic, as a major influence on their approach to antibiotic prescribing. Initially in the pandemic, to minimise face-to-face contact, participants started conducting almost all consultations remotely (mostly by telephone). Being unable to examine patients with RTI symptoms in person meant that clinical decisions were more difficult and felt more risky, and made GPs more likely to prescribe antibiotics to “*cover every eventuality*”. Participants interviewed in autumn 2020 wanted to return to in-person contact and examination.
“*I do feel that we probably have prescribed more than we normally would, because of the, especially at the beginning-in March, April, May, the lack of face-to-face consultations, or moving everything online. (…) if someone said they’d got a sore ear, and you can’t look at the ear, then you’re more likely to prescribe something, more likely than not, antibiotics. So that’s just one example of why I think we probably have “overprescribed” antibiotics more than usual.*”(GP10 November 2020)
“*What we absolutely need to do, as clinicians, is get our hands on the patients. It feels very wrong prescribing without actually listening to somebody’s chest. (…) so this is March, April, to really cover ourselves and cover the patient and make sure we’d covered every eventuality (…) we would be prescribing more readily than we would have before.*”(GP02 October 2020)

Most participants also described how communicating with patients, especially when not prescribing antibiotics, was more difficult and took longer in remote consultations because it required more detailed history-taking and safety-netting.
“*…we’re probably spending more time safety-netting than we used to because we haven’t laid our eyes on [patients] necessarily. So it’s more difficult to satisfy ourselves as clinicians that they don’t need antibiotics just based on the history. (…) Whereas in the past you just look at their ear and get all the information (…) Now you’re having to ask 20 questions to get the same information.*”(GP06 November 2020)
“*If you’ve looked in their ear, looked in their throat, listened to their chest, (…) and you can go through all the negative findings and say, look this is why I think it’s a viral infection. That conversation is a little bit more difficult if you haven’t seen them, and then they get that impression that, well you’re guessing at what’s going on.*”(GP18 June 2021)

Participants described introducing more face-to-face appointments from summer 2020, and by spring 2021 used a mix of remote and in-person consultations; for example, assessing the patient remotely and then making a decision about whether to see the patient in-person. This helped return to more “normal” antibiotic prescribing as they were less dependent on prescribing antibiotics to manage uncertainty and risk.
“*…the number of face-to-face consultations has definitely increased from the first wave. (…) …we are examining patients face-to-face when they need to be examined, like their ears and their throat and their chest etc., there should be far fewer so called inappropriate antibiotic prescribing…*”(GP10 April 2021)

Participants also reported sending patients to be seen in-person in COVID-19 clinics (“hot hubs”), which helped them avoid unnecessary antibiotic prescribing over the telephone.
“*Perhaps the fact that we’ve got the hot hub with the respiratory symptoms also helps us sort of, use that as a technique. (…) in a lot of situations we are able to say that we think that you need to be seen for that in which case to go to the hot hub. That kind of extra barrier, it does help us to reduce our antibiotic prescribing for respiratory symptoms and have those conversations around that and what’s normal, what to do if things get worse and so on.*”(GP11 November 2020)

Over time, improved access to COVID-19 testing and COVID-19 vaccinations allowed for increased in-person consultations as the risk of face-to-face contact was perceived as lower.

### 2.3. Changing Patient Presentations and Workloads

The number of patients and workloads, types of presentations in addition to patient behaviours and expectations also seemed to influence antibiotic prescribing throughout the pandemic. Early in the pandemic participants described general practice as quiet, with routine care reduced and far fewer patients than usual contacting general practice (e.g., due to concerns about risk or thinking that surgeries were closed). Participants reported significantly increasing demand and workloads in general practice since summer 2020 (“*an absolute deluge of work, just enormous quantities of work*” (GP06, November 2020)). This was described as due to the “*backlog*” of delayed issues, “*unmet need*”, more work “*spilling over*” from reduced secondary and specialist services, an increase in mental health issues and, later, delivering COVID-19 vaccinations. The changing patient numbers, together with shifting methods of consulting, seemed to influence the fluctuations in the number of antibiotics prescribed.
“*At the beginning, I would probably say we prescribed a lot less because we had so few contacts. People just weren’t contacting us for a start. I’d say once contact increased, then we probably prescribed more because we didn’t want to bring people in. Now most people are coming in, it’s probably reduced again.*”(GP15 May 2021)

The types of presentations also seemed to influence antibiotic prescribing. Initially, with the national lockdown, social distancing and school/nursery closures, participants reported having fewer patients with RTIs and, especially, far fewer children than typical. They accounted this to a lower spread of infections, better self-management and more patients with RTIs/COVID-19 being seen in hot hubs.
“*I have hardly seen any children at all since March which is a huge change as well. (…) I think a lot of it is because the majority of their infections are self-limiting illnesses (…) Also lots of the coughs, colds, fevers were being seen either in the COVID hubs or having COVID swabs and just staying at home, and so for that reason they didn’t really need to see a GP.*”(GP14 December 2020)

On the other hand, earlier in the pandemic participants reported being more likely to prescribe antibiotics for patients with RTI symptoms and (suspected) COVID-19. Antibiotics were perceived to be a safer option early in the pandemic due to uncertainties around distinguishing COVID-19 from a bacterial RTI, a lack of evidence on potential treatments for COVID-19 and COVID-19 guidelines (with antibiotics recommended in certain situations).
“*…our local guidelines on the treatment of COVID was, if the symptoms didn’t settle, then you would prescribe doxycycline so that was being prescribed.*”(GP02 October 2020)

Later on, a few participants reported reducing their prescribing for COVID-19 as evidence emerged showing that antibiotics are ineffective for it.

Finally, there appeared to be mixed views about the impact of the pandemic on patients’ behaviours and expectations for antibiotics. On the one hand, self-care and infection prevention behaviours might have improved due to the pandemic-related public health advice; on the other hand, there might have been little impact on patients’ attitudes to antibiotics as COVID-19 was a viral infection.
“*People have generally become more aware of infections and the way they spread and general things that they can do to try to avoid spreading of the infection, simple hygiene and so on. But I don’t necessarily think that people have become more aware of antibiotics as such because obviously this is a viral pandemic and it’s not a bacterial thing. I don’t think that antibiotic stewardship has been particularly high on the media agenda. This is not really coming across in the news to people that that’s an important thing to consider.*”(GP09 November 2020)

Some patients were perceived as presenting later (e.g., after waiting for COVID-19 test results), others as contacting the surgery earlier than pre-pandemic, at the first onset of symptoms (e.g., because of easier online contact).
“*Whereas previously trying to get hold of an appointment was difficult and therefore patient problems either self-resolved or they sought advice from elsewhere, for example seen a pharmacist, now they know they can just send in an e-Consult and then it’s done, so we’re getting patients presenting far, far earlier on…*”(GP17 June 2021)

Some participants also felt there was still patient preference for antibiotics (with patients described as “*a lot happier*” with “*a far lower threshold*” for antibiotics), increasing the pressure on GPs to prescribe antibiotics.
“*…lots of our patients don’t want to go into [town] to the COVID hubs. It’s a 45 min journey so they just don’t want to do it, so they do put us under quite a lot of pressure, I don’t want to go that far, can’t you just prescribe for me…*”(GP14 December 2020)

### 2.4. Changing Engagement with AMS Strategies

During the pandemic participants reported little engagement with AMS. Most reported a lack of discussions or initiatives related to antibiotics and AMS in practices, whereas a few described only discussing antibiotic prescribing in relation to COVID-19 and current guidelines. Participants also thought that the Clinical Commissioning Groups (CCGs: responsible for commissioning local health services, including medicines management and AMS) suspended their usual AMS work with practices. For example, participants reported not having or not recalling the usual meetings with CCG prescribing advisors and/or no feedback on their antibiotic prescribing data.
“*We had one meeting with a CCG pharmacist, (…) that would have been December and probably the next one was due in March or April and didn’t happen. And in the meantime, our entire lives have been consumed with COVID planning and getting through the work.*”(GP06 November 2020)

Participants expressed mixed views regarding whether and how different AMS strategies (that can be used to optimise antibiotic prescribing) were used and affected during the pandemic (Table 1). Some GPs described difficulties using particular strategies in remote consultations, e.g., communicating no need for antibiotics or an inability to use point-of-care C-reactive protein (CRP) testing. Others described adapting some strategies in remote consultations, e.g., sending links to written patient information leaflets or using clinical scores.

### 2.5. Shifting Priorities

Participants described how their priorities changed during the pandemic, which linked and overlapped with the changes described above. Early on, reducing COVID-19 transmission and protecting staff as well as patients took precedence. Over time, the priority and planning shifted to safely restoring face-to-face contact, adapting to new workflows (with remote and in-person appointments), dealing with the increasing demand/workloads and delivering vaccinations. Consequently, AMS was not perceived as a priority during the pandemic, even after the initial emergency period. It was perceived as less urgent, at least until the other priorities were addressed.
“*At the moment, [AMS] is not a priority. We’re overwhelmed with work and we’ve not got enough staff… And to do things proactively just now where they’re not directly impacting on our day to day work is really difficult to do. There are so many other priorities that we’ve got to try and keep the services running, and I suspect a lot of practices feel the same, that they’re struggling to keep services going, deliver the vaccine clinics, and deal with the unmet needs. And where something is not as immediate as that (…) there isn’t the capacity to do things… that don’t have that priority just now. And I’m afraid antibiotic prescribing is one of them. As is a whole lot of other prescribing to be honest with you.*”(GP08 May 2021)

While participants recognised the importance of AMS and re-engaging with it, they described no plans for it and that many uncertainties were impeding their planning. They suggested it would be useful for the CCGs to return to their usual AMS work (e.g., providing feedback on prescribing, incentives).
“*We don’t have any plans as a practice as such at the moment but we tend to respond very well to the [CCG] Medicines Optimisation Team’s strategies that they send out every now and then. So they run a prescribing quality scheme every year, which has actually been deferred this year because of COVID (…) there’s a possibility it won’t restart till next April but that tends to be the main driver for us in terms of affecting our prescribing habits.*”(GP09 May 2021)

Some planning seemed to relate to finding ways to manage the new ways of consulting and working as well as the increased demand and workloads. Telephone triage and remote consultations were perceived to be continued, and participants described the need to identify the optimal proportions and processes for remote and in-person consultations. A few participants reported expanding their teams with additional staff, especially non-GP clinicians (such as healthcare assistants (HCAs) or nurses whose qualifications in the UK involve clinical training but do not provide the right to prescribe medication), to help manage workloads and acute infections. Some described these changes as potential opportunities for improving the management of infections and workloads as well as antibiotic prescribing in the future, e.g., for triaging patients to acute infections teams (rather than directly to GPs) or for implementing point-of-care CRP testing.
“*Starting with telephone and video consultations opens up a much more, a much smoother pathway for doctors to start thinking about, and the [Healthcare Assistant] led step path after that so, i.e., we can have a quick conversation, take the history, take the concerns, take the ideas, and then tell people to come in and see someone, see an HCA for the necessary observations and tests.*”(GP11 November 2020)

## 3. Discussion

We found that GPs in England perceived and experienced a considerable and changing impact of the COVID-19 pandemic on how they managed infections, prescribed antibiotics and engaged with AMS. They perceived that their threshold for prescribing antibiotics (particularly for RTIs) decreased early in the pandemic, influenced by moving to remote consultations without physical examination and ambiguities related to treating patients with (suspected) COVID-19. Both factors increased GPs’ clinical uncertainty and perception that prescribing antibiotics was a safer option than not prescribing. While the likelihood of prescribing antibiotics for RTI symptoms seemed higher, fewer patients presenting in general practice meant that the overall number of prescriptions issued was not necessarily expected to be higher than pre-pandemic. Over the subsequent months, and especially after the lockdown, participants perceived their approach to antibiotic prescribing as returning to “normal” (pre-pandemic). This was linked to gradually increasing in-person consultations and examination, increasing patient numbers and emerging evidence that antibiotics offer no benefit for COVID-19 [32,33,34]. Prudent antibiotic prescribing and AMS were not considered a priority during the first year of the pandemic, even after the emergency phase, overshadowed by more urgent priorities. A return of CCG AMS work and considering opportunities for new ways of working and managing infections were suggested as part of future AMS.

Participants’ perceptions about the impact on antibiotic prescribing confirmed and added nuance to the analyses of prescribing data. We identified two distinct impacts early in the pandemic: (i) a greater tendency to prescribe antibiotics (mostly due to clinical uncertainties and the need for safety-netting), and (ii) fewer consultations and thus fewer prescriptions for RTIs (mostly due to reduced transmission and/or patients delaying or avoiding contact with general practices). Our findings are in line with an analysis showing that when taking into account the decreased number of appointments, the number of prescriptions was 6.7% higher than expected [18]. They are also supported by an analysis showing that March 2020 was associated with higher antibiotic prescribing followed by lower rates than predicted between April and August 2020 [20], while the incidence of RTIs decreased (particularly in April 2020) [20,35]. Although participants described having a “*lower threshold*” for prescribing antibiotics (which may suggest that some were of little/no benefit), it is possible that these prescriptions were appropriate and influenced by patients’ presentations (e.g., higher threshold for consulting, presenting later and access issues). It is also possible that too few antibiotics were prescribed (e.g., considering a peak in community-acquired bloodstream infections in May 2020 [36]). Therefore, these fluctuations (including the numbers of prescriptions and consultations) need to be considered when interpreting changes in antibiotic prescribing during the pandemic [19,21,22,23]. Investigating the appropriateness of antibiotic prescribing against guidelines, antibiotic prescribing in COVID-19/hot hubs and patients’ help-seeking as well as self-management behaviours would also provide helpful insights.

In this study we identified influences experienced by GPs to explain these fluctuations in prescribing. The major factor that seemed to increase the likelihood of prescribing early in the pandemic involved shifting to remote consultations and managing (new) risks to staff and patients. A mixed-methods study found a rapid move to 90% remote GP consultations by April 2020 (decreasing in the following months) and GPs’ concerns about clinical risk and safe thresholds for seeing patients face-to-face [37]. Similarly, a qualitative study across eight European countries, including the UK, found that changes in managing patients with RTIs, remote care and dealing with uncertainty among the most salient experiences of the transformation of primary care during the pandemic [38]. Pre-pandemic evidence on the impact of remote consultations on antibiotic prescribing in primary care is inconclusive [39]. Future research needs to consider the longer-term impact of remote consulting and the pandemic as related but also separate issues. We found that over time GPs shifted to using asynchronous and mixed-mode consultations (i.e., gathering and reviewing information on symptoms, a telephone consultation and a decision on whether a face-to-face appointment is needed)—this was perceived as helpful and enabling a more usual approach to prescribing.

We also found that dealing with an increasing workload was a major concern and priority, which additionally impeded engagement with AMS. Increasing consultations and other professional responsibilities leading to unmanageable workloads were a major issue in the UK’s general practice long before the pandemic [40,41], and insufficient time was perceived as one of the main barriers to CCG and general practice professionals engaging with AMS pre-pandemic [42]. In order to support prudent antibiotic prescribing and engagement with AMS, it is important to devise optimal ways of working as well as managing demand and workloads in primary care. These issues are also relevant to other types of prescribing and quality improvement.

During the COVID-19 pandemic GPs and CCGs focused on priorities seen as more urgent than AMS. As part of the STEP-UP study we also interviewed CCG professionals responsible for AMS in general practices, who also reported decreased focus on AMS during the pandemic and intentions to re-prioritise AMS (unpublished data). Similarly, there was a negative impact of COVID-19 on routine AMS activities in hospitals [9]. Pre-pandemic (and as found in this study), AMS in general practices largely depended on the engagement of CCG prescribing advisors, mostly with strategies such as prescribing targets, feedback and incentives [42]. Our participants suggested that the re-engagement of CCGs with, and return to, promoting/use of AMS strategies (starting with audits and feedback) is essential to bring antibiotic prescribing into focus again.

Existing AMS strategies can be adapted to, or repurposed for, use in the context of COVID-19 [12]. Currently, we need to better understand how AMS strategies may fit in, or be adapted to, post-pandemic primary care. Some GPs perceived the pandemic as creating opportunities to rethink and reorganise the management of acute infections in primary care. This may help in considering AMS a part of processes for improving infection management, patient outcomes and safety, rather than as a separate initiative. New approaches might involve, e.g., more non-GP clinicians and acute/minor infections teams as the first or primary point of contact for non-vulnerable patients with acute infections, or a wider use of point-of-care testing for RTIs to guide antibiotic prescribing [43,44].

### 3.1. Limitations

Due to the rapid nature of the pandemic, we adapted and extended an existing study to explore the impact of COVID-19 on AMS. This led to some limitations. To ensure prompt recruitment, we used convenience sampling, including participants in the STEP-UP study (focused on the implementation of AMS strategies) and others who expressed interest. Although the views of participants from the STEP-UP study and additional GPs did not seem to differ, participants’ views might have still differed from non-participants. Our aim was to capture change over time; however, most change took place at the start of the pandemic before we conducted the first round of interviews. Thus, the findings related to the initial pandemic phase rely on recollections and may be subject to recall bias and reinterpretation. Most participants were from one Clinical Research Network area; GPs in other areas of England or countries might have reported different views and experiences. While we deemed the data from the 24 interviews as sufficient to answer the research question and develop rich themes, there is always a possibility that further interviews could have provided more nuanced data and answered additional questions (e.g., regarding potential differences in perceptions and experiences between participants from different areas or types of practices). As with all qualitative studies, the findings are time- and context-specific.

### 3.2. Implications

The main implications of this study relate to the need to resume AMS activities in primary care and adapt them to post-pandemic models of care. Prioritising AMS requires addressing other issues, such as finding ways to manage the increased workloads or appropriate mix of remote and in-person consultations. Most immediately, CCGs need to resume AMS work to drive re-focusing on, and reinforcing, the importance of prudent antibiotic prescribing and AMS; for example, reviving prescribing data feedback, audits and meetings to discuss prescribing data and targets. In the long term AMS will need to fit within post-pandemic primary care models/delivery. This will likely involve much higher levels of triage and non-face-to-face consulting than pre-pandemic, including a mix of asynchronous (e-consults) and synchronous (telephone/video calls) consulting. It might also involve new workflows and health professionals. Thus, we need to exploit the learning from the pandemic and identify further useful adaptations. For example, we learnt that some AMS strategies can be slightly adapted to be used with triage as well as remotely (e.g., patient information leaflets, clinical scores and delayed/back-up prescriptions). Our findings also highlighted opportunities, created by the disruption of the pandemic, for improving the management of acute infections. For example, opportunities to rethink and reorganise workflows (e.g., triage, mixed-mode consultations) in addition to potential/planned increases in team-based approaches, e.g., with the new models of patients triaged to be first consulted by non-GP clinicians, including clinically trained non-prescribers (e.g., HCAs, nurses, paramedics and physician associates) in minor/acute infection teams before deciding whether there is a need for the patient to be seen by a GP. Such changes will also require more focus on consistent, practice-based approaches to managing infections and prescribing. Other opportunities may involve the promotion of infection prevention measures, self-care and pharmacy-based services (e.g., with advice, patient information leaflets and/or point-of-care testing). More research needs to identify how to best implement different AMS strategies and optimally manage acute infections in the post-pandemic context. It also needs to identify the longer-term impact of remote/mixed-mode consulting on infection management and prescribing thresholds. A better understanding of (changing) patients’ help-seeking and self-management behaviours as well as their views on antibiotics (during and especially as we are coming out of the pandemic) is also important.

## 4. Methods

This was a qualitative interview study as a qualitative methodology is best suited to explore the experiences of the target group and the question of *why* changes were perceived to occur. We took a longitudinal approach to data collection (two time points) in addition to a thematic and longitudinal approach to analysis. We followed the Standards for Reporting Qualitative Research [45].

### 4.1. Study Participants

We used a convenience approach for recruitment and sampling. Firstly, to quickly identify participants we included GPs participating in the on-going study on implementing evidence-based AMS strategies (the STEP-UP study [46]). Then, we recruited additional GPs through the local Clinical Research Network. Participants were from the West Midlands as well as Thames Valley and South Midlands areas in England. All participants provided informed consent verbally and written records of consent were made. Participation was reimbursed.

### 4.2. Data Collection

Semi-structured interviews were conducted by telephone or video calls (depending on participants’ preferences) by an experienced qualitative, non-clinical researcher (AJB). In order to capture potential changes over time we took a longitudinal approach (cross-sectional type, as described in [47]), with interviews conducted at two time points: (i) autumn (late October to early December) 2020, and (ii) spring (late April to early June) 2021. Three interview guides were used (Appendix A): (i) for GPs in the STEP-UP study in autumn 2020 (with questions about the STEP-UP AMS initiatives and the impact of COVID-19); (ii) for additional GPs in autumn 2020 (about the impact of COVID-19 on antibiotic prescribing and AMS since the start of the pandemic); and (iii) for interviews in spring 2021 (about the impact of COVID-19 on antibiotic prescribing and AMS during the year of the pandemic, as well as future priorities and plans). The interviews were audio-recorded, transcribed verbatim and the transcripts were checked as well as anonymised.

### 4.3. Data Analysis

We took a longitudinal approach to analysis, with the whole sample analysed together with attention to changes and consistencies over time [48]. Notes were taken during and summaries were made following each interview; they helped with data familiarisation and initial ideas about emerging findings. All transcripts were uploaded to qualitative data management software (NVivo, v.12, QRS International) and analysed thematically (taking an inductive and realist/essentialist approach) [33]. Two researchers (A.J.B. and K.M.) initially independently inductively coded four transcripts (two each from autumn and spring) and then compared and integrated their codes. Using the notes from all interviews and team discussions, higher-level categories and themes were identified, resulting in a hierarchical coding framework. The coding framework was used to code the remaining transcripts (K.M. coded 15, A.J.B. 12 transcripts) while codes were added to capture new details/data. All coding was reviewed for consistency. Detailed notes (“audit trail”) and a team-based approach (“researcher triangulation”) were two main ways to ensure the trustworthiness of the analysis.

## Figures and Tables

**Table 1 antibiotics-10-01531-t001:** Views about the use of, and impact on, AMS strategies during the COVID-19 pandemic.

AMS Strategy	Summary of Views
Communication strategies to explain prescribing decisions	There were mixed views: Some participants described no change in their communication with patients during the pandemic and remote consultations. Others described that moving to remote consultations made them more focused on detailed history-taking and safety-netting due to the lack of physical examination, and more focused on discussing patient concerns, expectations and understandings of normal/concerning symptoms.
Delayed/back-up prescriptions	There were mixed views: Some participants reported no change in their use of delayed/back-up prescriptions compared to pre-pandemic (e.g., whether and when they used them, or not used them). Some participants reported using delayed/back-up prescriptions less often, in favour of prescribing immediate antibiotics. Others reported using more delayed/back-up prescriptions during the pandemic as a safety net (especially with limited options to see the patient) or to limit re-consultations. Participants described shifting to fully electronic prescribing (before and some since the pandemic); few perceived electronically sent delayed/back-up prescriptions to be more difficult for patients to understand (e.g., rather than a post-dated paper prescription).
Point-of-care C-reactive protein (CRP) testing	Participants reported not being able to use the point-of-care testing equipment due to the lack of, and limited, face-to-face contact. Some thought that it could/might be useful for face-to-face consultations, e.g., when they become safer and more frequent again. Some speculated how point-of-care CRP testing could be implemented in the future, e.g., in pharmacies, by triaging patients to have the test and by non-GP clinicians.
Written patient information leaflets	Most participants reported an increased sharing of patient information due to being able to electronically send or text message links to patient information leaflets/websites to patients, and becoming more used to this system function. Some reported not giving out (printed) information leaflets due to the lack of/limited in-person contact.
Guidelines	Participants reported following prescribing guidelines as a strategy to ensure appropriate antibiotic prescribing, and many reported that this remained unchanged during the pandemic. Some referred to following COVID-19 guidelines in relation to deciding about antibiotics for patients with (suspected) COVID-19.
Clinical scores or templates (e.g., Centor [30], FeverPAIN [31])	Some reported using and adapting clinical scores/templates for new ways of consulting during the pandemic, e.g.,: Using algorithms as part of the triage process (e.g., to identify potential infections, sepsis or COVID-19). Using FeverPAIN and/or Centor in remote consultations as part of an assessment with slight adaptations, e.g., instead of making observations in a physical examination, they would ask patients about these (e.g., fever, swollen glands), ask to see their throat in a video or picture or ask a relative to look at the patient’s throat and describe it.

## Data Availability

The data presented in this study are available on request from the corresponding author. Additional data (quotes) supporting the findings are available in Appendix A.

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
