# Peer review of "Impact of the COVID-19 Pandemic on Community Antibiotic Prescribing and Stewardship: A Qualitative Interview Study with General Practitioners in England"

_antibiotics, 2021, doi:10.3390/antibiotics10121531_

Round 1

Reviewer 1 Report

Dear authors,

I applaud your efforts with this research study as the topic is important and can contribute to the general knowledge. The methods are appropriate and designed to address the question being asked. As with all qualitative research, perceptions are time and context sensitive. The representativeness of study sample to the population in question would add robustness to the resulting data. Information about the study sample need to be presented her, the extent of their geographic service, number of patients seen per time compared to the population average, the nature of locations they service (urban vs rural),types of patients they see,....

Finally, how was the decision made that the sample size is sufficient?

Best,

Author Response

  1. I applaud your efforts with this research study as the topic is important and can contribute to the general knowledge. The methods are appropriate and designed to address the question being asked. As with all qualitative research, perceptions are time and context sensitive.

Thank you for this positive comment.

  1. The representativeness of study sample to the population in question would add robustness to the resulting data. Information about the study sample need to be presented here, the extent of their geographic service, number of patients seen per time compared to the population average, the nature of locations they service (urban vs rural),types of patients they see,....

As reported in the Methods and Limitations (lines 97 and 503), we used a common ‘convenience approach to recruitment and sampling’ due to the rapid nature of recruitment and adapting the existing study for the new research question. This is standard practice for many qualitative studies as the purpose is to gain insight into views and experiences of individuals in context rather than produce generalisations about a population. We acknowledge the limitations of this approach in the ‘Limitations’ section. As reported, participants were from the West Midlands and Thames Valley and South Midlands areas in England. We do not have the data on the number and types of patients seen.

We interviewed 18 GPs for this study and it would not be feasible to ensure a sample representative of all UK GPs and according to the diverse types of characteristics mentioned by the reviewer. Representativeness and generalisability are concepts used to assess the quality of quantitative studies, and they do not apply to qualitative methodologies in a similarly straightforward way – there are long-standing debates about these concepts and quality criteria between quantitative and qualitative methodologies. Our aim was to explore and share experiences of some GPs, rather than identify their prevalence and moderators (for which quantitative methods are more suitable).

  1. Finally, how was the decision made that the sample size is sufficient?

As reported in the Results on lines 136-7, we stopped recruitment when we considered the data sufficient to answer the research question. By that we meant that we had enough data related to the research question to develop rich categories and themes (i.e. supported by multiple quotes from multiple participants), and that conducting more interviews would bring diminishing returns. We indirectly referred to the concept of data saturation, defined as ‘information redundancy or the point at which no new themes or codes “emerge” from data’ (Braun and Clarle, 2021; https://doi.org/10.1080/2159676X.2019.1704846). However, we avoided using the term ‘data saturation’ due to the debates within qualitative methodology research on the value, usefulness and definitions of the concept (as discussed in the paper by Braun and Clark above, and many others). From our research experience and other qualitative studies on antibiotic prescribing and stewardship in primary care, most studies involve samples of nearly 20 participants or interviews, which is similar to our 18 GPs and 24 interviews. Longitudinal qualitative studies often involve smaller numbers of participants as these are followed over time.

We have added a clarification on lines 137-9 and in ‘Limitations’ on lines 512-516.

Reviewer 2 Report

Dear authors

although the topic of your paper is quite interesting as it considered perceptions and behaviors of GPs, I honestly think that the results are not so scientifically sound given the methodology used, so that the paper seems not so suitable for publication in Antibiotics.

Best regards 

Author Response

  1. Although the topic of your paper is quite interesting as it considered perceptions and behaviors of GPs, I honestly think that the results are not so scientifically sound given the methodology used, so that the paper seems not so suitable for publication in Antibiotics.

We were surprised to read this comment – the reviewer seems to be indicating that because this is a qualitative study it is not scientifically sound. We would refer the reviewer to a widely cited open letter on the topic (on the role and value of qualitative research) from 2016 in the BMJ: https://doi.org/10.1136/bmj.i563. We also note that qualitative studies are regularly published in Antibiotics and that it is common to see methodologically similar papers, based on interviews and focus groups.

Reviewer 3 Report

The authors describe a qualitative analysis of antibiotic use and AMS practices among primary care providers in England relative to the CoVID-19 pandemic. The manuscript provides an interesting insight into prescriber attitudes and beliefs about antibiotic prescribing in the context of mandatory virtual appointments made necessary by a global pandemic.

Major comments:

  1. Abstract: Line 14-43- The abstract conclusion “While the pandemic disrupted the usual ways of working, it also produced opportunities” seems to be a minor theme of this manuscript (really only mentioned in 346-355). Would suggest removing this as a conclusion, or adding more evidence to support the claim.
  2. Results: Line 92- I question why the authors felt that 18 GP interviews was sufficient to answer the research question. Surely this sample is not representative of all GP practices in England? Perhaps the manuscript should simply omit this justification and be more transparent about the sample that it currently represents.
  3. Results: Lines 242-254- this section seems like it would fit better in the section about fluctuations in antibiotic use, not changing patient presentations and workload.
  4. Implications: Lines 465-470- The statement “we learnt that some AMS strategies can be slightly adapted…” and the sentences about the management of acute infections do not seem to have been adequately described in the body of the manuscript. Furthermore, the quote in Line 352 about the Healthcare Assistant may not be clear to readers outside of the UK. As a US reader, I am unsure about that an HCA model would mean and how this person would be qualified to administer tests.

Minor comments:

  1. Introduction: The statement “antimicrobial stewardship and optimizing antibiotic prescribing and use” seems odd, because isn’t antimicrobial stewardship by definition a practice that optimizes antibiotic prescribing and use? Why is there an “and”?
  2. Results line 119-122- Would suggest that ‘ambiguity’ is not the most accurate word here. It sounds as though the authors are trying to suggest that the surveyed practitioners did not accurately/confidently predict how 2020 prescribing compared to previous years. It may be more valuable to describe this in the context of what the prescribing actually was.
  3. It’s not clear to me as a practitioner who does not live in England what is meant by “patient leaflets” and how that might differ from “communication strategies to explain prescribing decisions”
  4. I suspect that readers who do not practice in ambulatory/primary care may not know these tools (Feverpain, Centor) -is it necessary to refer to them by name?

Author Response

  1. The authors describe a qualitative analysis of antibiotic use and AMS practices among primary care providers in England relative to the CoVID-19 pandemic. The manuscript provides an interesting insight into prescriber attitudes and beliefs about antibiotic prescribing in the context of mandatory virtual appointments made necessary by a global pandemic.

Thank you for this positive comment.

  1. Abstract: Line 14-43- The abstract conclusion “While the pandemic disrupted the usual ways of working, it also produced opportunities” seems to be a minor theme of this manuscript (really only mentioned in 346-355). Would suggest removing this as a conclusion, or adding more evidence to support the claim.

Although this is briefly reported in the results as it was part of the theme, we consider it an important finding and implication of this study.

We have rephrased the abstract (lines 43-45) to clarify this is an implication rather than conclusion.

  1. Results: Line 92- I question why the authors felt that 18 GP interviews was sufficient to answer the research question. Surely this sample is not representative of all GP practices in England? Perhaps the manuscript should simply omit this justification and be more transparent about the sample that it currently represents.

Please see our response to comment #3 by Reviewer 1. We do not present the sample as one which is representative of all practices in England; this would be inappropriate in qualitative research. Qualitative research seeks to describe its sample so readers can assess the transferability of findings to other populations.

  1. Results: Lines 242-254- this section seems like it would fit better in the section about fluctuations in antibiotic use, not changing patient presentations and workload.

The comment refers to the findings on lines 245-257 in the revised manuscript. These findings are that patients were sent to COVID-19 clinics and that over time improved access to testing and vaccines increased in-person consultations; both of these factors influenced antibiotic prescribing (i.e. increasing in-person consultations helped avoid unnecessary antibiotics). We consider these as examples of the theme ‘changing ways of working and consulting’ as they refer to whether and how patients were consulted in person, and how these influenced the fluctuations in antibiotic prescribing. The theme ‘fluctuations in antibiotic prescribing’ is an overarching theme, which is illustrated by themes 2-4. Hence, we think that the reported findings fit better in the section that they were reported in.

  1. Implications: Lines 465-470- The statement “we learnt that some AMS strategies can be slightly adapted…” and the sentences about the management of acute infections do not seem to have been adequately described in the body of the manuscript. Furthermore, the quote in Line 352 about the Healthcare Assistant may not be clear to readers outside of the UK. As a US reader, I am unsure about that an HCA model would mean and how this person would be qualified to administer tests.

In the revised manuscript, the referred quote starts on line 407 and the referred statement starts on line 530.

The main focus of the study (and the research question) focused on the impact of the COVID-19 pandemic on antibiotic prescribing and stewardship. Participants were only asked about current and future priorities and about views on potential / hypothetical longer-term impacts in the later interviews (i.e. this topic was not explored in the first round of interviews). As reported in the findings, some participants described an uncertainty about the future, even near future, and challenges to planning because of the nature of the situation. A few described some potential future changes in how acute infections are managed that may be needed (e.g. to manage increasing demand/workload) and may be possible given the changes caused by the pandemic. These findings are given proportional attention and reporting in the manuscript alongside other findings.

For international readers, we have clarified the qualifications of the HCA in the results on lines 400-1 and what the new models involving non-GP clinicians might be on lines 536-540.

  1. Introduction: The statement “antimicrobial stewardship and optimizing antibiotic prescribing and use” seems odd, because isn’t antimicrobial stewardship by definition a practice that optimizes antibiotic prescribing and use? Why is there an “and”?

Thank you - we have amended the sentence (line 51).

  1. Results line 119-122- Would suggest that ‘ambiguity’ is not the most accurate word here. It sounds as though the authors are trying to suggest that the surveyed practitioners did not accurately/confidently predict how 2020 prescribing compared to previous years. It may be more valuable to describe this in the context of what the prescribing actually was.

Thank you - we have amended the sentence to clarify what we meant (lines 166-8).

  1. It’s not clear to me as a practitioner who does not live in England what is meant by “patient leaflets” and how that might differ from “communication strategies to explain prescribing decisions”

Thank you for the opportunity to clarify this. Patient leaflets are written leaflets with relevant information (e.g. about self-care) for patients that can be given to patients either as printed leaflets or sent to them electronically. These are often used to support and reinforce verbal communication strategies. In systematic reviews (e.g. de Bont et al. 2015 BMJ Open, Tonkin-Crine et al. 2017 Cochrane, Sustersic et al. 2017 Health Expectations) these are referred to as ‘patient information leaflets’ and in some other literature (e.g. O’Sullivan et al. 2016 Cochrane) as ‘written information for patients’).

In the revised manuscript, we have changed the phrase to ‘written patient information leaflets’ on line 355, in Table 1, and on line 532.

  1. I suspect that readers who do not practice in ambulatory/primary care may not know these tools (Feverpain, Centor) -is it necessary to refer to them by name?

We only refer to FeverPAIN and Centor scores in Table 1 as examples of scores mentioned by our participants. We think that there is no harm in giving the names as examples.

In Table 1 in the revised manuscript, we have added references numbers 34 and 35 (highlighted in the list of references) to these two scores for readers who want to look them up.

Round 2

Reviewer 2 Report

Dear authors

I apologize if my comments have given you the impression that I undervalued scientifically your study. Yes you are right qualitative research studies are gaining more and more importance and consideration. The sentences you added in the limitation sections have mitigated your conclusions, as my real concern is the generalizability of your results considering the small sample size and the fact that the enrolled GPs were already participating to a study promoting prudent antibiotic use (International Journal of Nursing Studies 47 (2010) 1451–1458)